# Prognostic Significance of Endocrine-Related Adverse Events in Patients with Melanoma, Non-Small Cell Lung Cancer and Urothelial Cancer After Treatment with Immune Checkpoint Inhibitors: A Systematic Review and Meta-Analysis

**DOI:** 10.3390/cancers17223675

**Published:** 2025-11-17

**Authors:** Stylianos Kopanos, Charalampos Filippatos, Pantelis Rousakis, Ioannis V. Kostopoulos, Constantin N. Baxevanis, Anastasios Tentolouris, Maria Gavriatopoulou, Ourania Tsitsilonis, Ioannis Ntanasis-Stathopoulos

**Affiliations:** 1Academic Department of Endocrinology, Diabetes and Infectiology, Klinikum Bielefeld, Medical School and University Medical Centre East Westphalia-Lippe, Bielefeld University, 33615 Bielefeld, Germany; 2Department of Clinical Therapeutics, School of Medicine, National and Kapodistrian University of Athens, 157 72 Athens, Greecemgavria@med.uoa.gr (M.G.); johnntanasis@med.uoa.gr (I.N.-S.); 3Flow Cytometry Unit, Department of Biology, School of Science, National and Kapodistrian University of Athens, 157 72 Athens, Greece; prousakis@biol.uoa.gr (P.R.); ivkostop@biol.uoa.gr (I.V.K.); cnbaxevanis@biol.uoa.gr (C.N.B.); rtsitsil@biol.uoa.gr (O.T.); 4Department of Propaedeutic Internal Medicine, ‘Laiko’ General Hospital, Medical School, National and Kapodistrian University of Athens, 157 72 Athens, Greece; antentol@med.uoa.gr

**Keywords:** immune checkpoint inhibitors, melanoma, non-small cell lung cancer, urothelial cancer, endocrinopathies

## Abstract

Immune checkpoint inhibitors have transformed cancer therapy by enhancing the body’s own immune response against tumors. However, this immune activation can also affect hormone-producing glands, leading to thyroid, pituitary, pancreatic, or adrenal dysfunctions. These immune-related endocrine effects may serve as signals of an active immune system, but their true relationship to treatment success remains uncertain. In this systematic review and meta-analysis, including 43 studies and more than 17,000 patients with melanoma, lung, and bladder cancer, we found that individuals who developed endocrine side effects generally lived longer. Yet, further analyses suggested that treatment benefit and side effects may occur independently. These findings highlight the need for careful monitoring of hormonal changes during immunotherapy and suggest that endocrine complications could reflect immune activation rather than directly predict therapy effectiveness.

## 1. Introduction

Malignant melanoma, non-small cell lung cancer (NSCLC), and urothelial carcinoma remain among the leading causes of cancer-related mortality worldwide. Cigarette smoking, ultraviolet radiation, and chronic inflammation are well-established risk factors contributing to the development of these malignancies [1,2]. Despite advances in targeted therapies and platinum-based chemotherapy, the five-year survival rates for advanced or metastatic disease historically remained below 30–40% [3]. Immune checkpoint inhibitors (ICIs) have transformed this landscape by restoring antitumor immunity through the inhibition of immune-regulatory pathways. Under physiological conditions, cytotoxic T-lymphocyte–associated antigen 4 (CTLA-4), programmed cell death protein-1 (PD-1), and programmed death-ligand 1 (PD-L1) act as co-inhibitory receptors expressed on activated T cells, downregulating immune responses to maintain tolerance [4,5,6].

The immune system plays a critical role in recognizing and eliminating cancer cells through the activation of antitumor-reactive T cells via major histocompatibility complex (MHC)-mediated tumor (neo)antigen presentation by antigen-presenting cells. Co-stimulatory signals, such as CD28 binding to CD80/CD86, further enhance T-cell activation [7,8], whereas inhibitory molecules, including CTLA-4 and PD-1, downregulate immune responses to prevent autoimmunity and maintain immune homeostasis [9]. CTLA-4 is expressed on activated T cells and binds to CD80/CD86 with higher affinity than CD28, suppressing T-cell activation. Similarly, PD-1, also expressed on T cells, binds PD-L1/PD-L2, leading to T-cell exhaustion and apoptosis [10,11].

Tumors exploit these pathways to evade immune surveillance by creating an immunosuppressive tumor microenvironment (TME), inhibiting tumor-specific T-cell responses. ICIs, i.e., monoclonal antibodies targeting CTLA-4, PD-1, and PD-L1, disrupt these inhibitory mechanisms, enhancing T-cell functions and antitumor immunity, and have been approved for an expanding range of malignancies. Nevertheless, ICIs also activate autoreactive T cells and predispose patients to autoimmunity and the development of immune-related adverse events (irAEs), such as autoimmune diabetes or thyroiditis [5,10,12].

Endocrine irAEs occur in approximately 10–20% of patients receiving ICIs, although reported rates vary widely depending on the intensity of biochemical monitoring and the specific ICI class. Thyroid dysfunctions are the most frequent events, whereas hypophysitis and adrenal insufficiency occur less often and are strongly associated with anti–CTLA-4 therapy. These frequencies should be interpreted with caution, given that routine pituitary or adrenal function testing is not universally performed in clinical trials.

Endocrine irAEs typically manifest 7–20 weeks after ICI-treatment initiation. Their pathophysiology is thought to involve the activation of autoreactive T cells and the production of self-targeting autoantibodies. While hormonal replacement can manage these conditions, the underlying glandular dysfunction is often irreversible [13,14]. Anti-CTLA-4 therapies mobilize naïve T cells, which initiate immune responses against the pituitary gland (hypophysitis), leading to immune cell infiltration and glandular inflammation, further causing structural damage and symptoms such as headaches, visual impairment, and hormone deficiencies [15]. Thyroid irAEs are more frequently associated with anti-PD-1/PD-L1 agents, which primarily activate exhausted effector T cells, potentially due to higher PD-L1/PD-L2 receptor density in the thyroid gland [16]. Thyroid dysfunctions are the most common endocrine irAEs, ranging from transient hyperthyroidism to chronic hypothyroidism, and are often asymptomatic, detected through routine biochemical screening. In some cases, thyroiditis presents with symptoms such as tachycardia and palpitations, while symptomatic hypothyroidism may necessitate therapy discontinuation. Thus, regular thyroid function assessment is recommended throughout ICI treatment [17].

Although rare, checkpoint inhibitor–associated autoimmune diabetes mellitus (CIADM) and adrenal insufficiency have been increasingly associated with anti-PD-1/PD-L1 therapies. CIADM (also termed ICI-DM, CPI-DM) is thought to occur secondary to autoimmune destruction of pancreatic β-cells. According to the American Diabetes Association, this condition is characterized by a rapid onset of insulin-dependent diabetes, often presenting with diabetic ketoacidosis and low or undetectable C-peptide levels, indicating near-complete loss of endogenous β-cell function. Notably, fewer than half of affected individuals have the typical autoantibodies seen in classic type 1 diabetes, suggesting a distinct pathobiology from traditional autoimmune diabetes [18,19]. Adrenal insufficiency, characterized by enlarged hypermetabolic adrenal glands, may manifest subclinically or with symptoms like hyponatremia. Early detection is performed by cortisol and ACTH level assessment [20].

Emerging evidence suggests that the development of all the aforementioned endocrine irAEs correlates with improved survival outcomes, making their presence a potential biomarker for ICI treatment efficacy. These endocrine irAEs can mirror robust immune activation and may coincide with improved clinical outcomes, yet their mechanistic and prognostic significance remains incompletely understood.

The present systematic review and meta-analysis synthesizes current evidence on endocrine irAEs across melanoma, NSCLC, and urothelial carcinoma, integrating data from 43 studies and over 17,000 patients. To our knowledge, this is the first study to explore the incidence, timing, and prognostic relevance of endocrine irAEs across these cancer types using meta-regression models to assess their potential as biomarkers of immunotherapy efficacy.

## 2. Materials and Methods

### 2.1. Study Design and Objective

This systematic review and meta-analysis analyzed 43 studies to evaluate the association between ICIs and endocrine irAEs in three cancer types: melanoma, NSCLC, and urothelial cancer. It also examined the clinical impact of these irAEs. The analysis incorporated randomized controlled trials (RCTs), cluster RCTs, interrupted time series (ITS) studies with at least three pre- and post-intervention data points, and controlled before–after (CBA) studies with a 3–4 months’ washout period.

### 2.2. Search Strategy and Data Sources

A comprehensive literature search was conducted in PubMed, Embase, Cochrane CENTRAL, Web of Science, and Scopus to identify relevant studies evaluating ICIs and endocrine irAEs in melanoma, NSCLC, and urothelial carcinoma. Manual searches of references and databases, including metaRegister, PROSPERO, FDA, ClinicalTrials.gov, and the WHO International Clinical Trials Registry Platform, were performed to identify additional studies. Conference abstracts from the American Society of Clinical Oncology, European Society of Oncology, and American Association of Endocrinology were reviewed. The search spanned studies published from 1 January 2012 to 20 May 2025, with no language restrictions applied. The search strategy Mendeley Mendeleyincorporated MeSH terms and free-text keywords, covering terms related to ICIs (e.g., PD-1, PD-L1, CTLA-4 inhibitors), endocrine toxicities (e.g., thyroid dysfunction, hypophysitis, adrenal insufficiency, diabetes mellitus), and tumor types of interest. Boolean operators (AND/OR) were used to refine results. The full search string used for PubMed is as follows:

(“immune checkpoint inhibitors” [MeSH Terms] OR “checkpoint blockade” OR “PD-1 inhibitor” OR “PD-L1 inhibitor” OR “CTLA-4 inhibitor” OR “pembrolizumab” OR “nivolumab” OR “atezolizumab” OR “ipilimumab” OR “durvalumab” OR “immune therapy”)

AND (“immune-related adverse events” [MeSH Terms] OR “endocrine toxicity” OR “endocrinopathy” OR “thyroid dysfunction” OR “hypophysitis” OR “adrenal insufficiency” OR “diabetes mellitus” OR “thyroiditis”)

AND (“melanoma” [MeSH Terms] OR “non-small cell lung cancer” OR “NSCLC” OR “urothelial carcinoma” OR “bladder cancer”)

AND (“clinical trial” [MeSH Terms] OR “randomized controlled trial” OR “RCT” OR “cohort study” OR “prospective study”)].

Equivalent strategies were adapted for other databases.

### 2.3. Eligibility Criteria

Studies between 2012 and 2025 on patients with moderate symptoms in remission maintenance were eligible. Inclusion criteria were defined according to the PICO framework (Table 1). We included randomized controlled trials, cohort studies, and controlled before–after studies that met the following criteria: (1) adult patients (≥18 years) with histologically confirmed melanoma, NSCLC, or urothelial carcinoma; (2) treatment with at least one ICI (anti–PD-1, anti–PD-L1, or anti–CTLA-4 therapy) as monotherapy or in combination; (3) reporting of endocrine irAEs such as thyroid dysfunction, hypophysitis, adrenal insufficiency, or immune-mediated diabetes; and (4) availability of survival outcomes, specifically progression-free survival (PFS) and/or overall survival (OS), or sufficient data to calculate hazard ratios (HRs) or odds ratios (ORs). Both prospective and retrospective studies were eligible if endocrine irAEs were explicitly described and confirmed by clinical or biochemical evaluation.

Relevant biomarkers [e.g., lipid profile, thyroid antibodies, Homeostatic Model Assessment for Insulin Resistance (HOMA-IR), hemoglobin A1c (HbA1c)] and hormonal concentrations [e.g., follicle-stimulating hormone (FSH), luteinizing hormone (LH), corticotropin-releasing hormone (CRH)] were also assessed.

Exclusion criteria included: (1) studies on cancers other than melanoma, NSCLC, or urothelial cancer, (2) trials with focus on outcomes unrelated to PFS, OS, or endocrine irAEs, reference to conventional chemotherapy or radiotherapy-only treatments, (3) non-English articles and case reports, editorials, or (4) studies lacking free access to full text or with insufficient data for quantitative synthesis. Excluded were studies focusing on other organ systems and presenting mild acute symptoms. Cross-sectional studies, case–control trials, and case reports were also omitted.

Ethical approval and patient consent were not required for this study, as the analysis was based exclusively on aggregated data from previously published studies available in the public domain. This systematic review and meta-analysis was conducted in accordance with the Preferred Reporting Items for Systematic Reviews and Meta-Analyses (PRISMA) guidelines, and the review protocol was prospectively registered in PROSPERO (Registration ID: CRD42025646504) [21].

### 2.4. Data Selection and Analysis

Search results were imported into Mendeley (https://www.mendeley.com/ (accessed 15 November 2025)) for de-duplication and managed using the Covidence software (https://www.covidence.org (accessed 15 November 2025)). Titles and abstracts were screened against inclusion criteria without blinding. Relevant full texts were reviewed to retain the most scientifically rigorous studies. Data extraction forms in Excel captured trial characteristics, participant demographics, interventions, and outcomes, including endocrine biomarkers and clinical manifestations (Appendix A).

Data from parallel group studies were synthesized narratively and statistically using the RevMan software (Version: 7.2.0 (2024)). Continuous outcomes were analyzed using weighted mean differences with 95% confidence intervals, while dichotomous outcomes were evaluated with odds ratios (ORs) using random-effects models to reduce variability. The PRISMA Flowchart is depicted in Figure 1.

Subgroup analyses were conducted based on: (1) the type of endocrine irAE (e.g., thyroiditis, hypophysitis); (2) diabetes status (type 1 vs. type 2 vs. non-diabetics); (3) cancer type (melanoma, NSCLC, urothelial cancer); (4) agent type (PD-1/PD-L1 vs. CTLA-4 inhibitors); and (5) bias risk (low/moderate vs. high). This methodological approach ensured robust data collection and analysis, enabling a comprehensive evaluation of ICIs’ impact on endocrine dysfunctions.

### 2.5. Statistical Analysis

Heterogeneity was assessed using the Q tests and I^2^ statistic, the latter quantifying the proportion of variability that cannot be attributed to random error. An I^2^ < 40% indicates low heterogeneity, whereas an I^2^ > 80% indicates substantial heterogeneity [22]. We also examined the between-study variance τ^2^. Random effect models were used to calculate pooled effect estimates. Cumulative forest plots were used to represent interactive effect size estimation. Funnel plots and Egger’s test were used for the assessment of publication bias [23].

We conducted both univariate and multivariate meta-regression analyses using random-effects models (REML estimator) to explore whether the log-transformed hazard ratios (logHR) for PFS were associated with the log-transformed OR (logOR) for irAEs. The models also assessed the moderating influence of cancer type (melanoma, NSCLC, urothelial carcinoma), adverse event severity (any vs. grade 3–4), and drug comparison (anti-PD-1 vs. anti-PD-L1, nivolumab vs. pembrolizumab). Residual heterogeneity and explained variance (R^2^) were reported.

We also conducted univariate and multivariate random-effects meta-regression analyses using study-level variables (age, median follow-up, prior medical treatment, etc.) to explore potential associations between treatment efficacy (log-transformed hazard ratios for PFS or OS) and toxicity (log-transformed ORs for adverse events). Adjustment for study-level covariates was limited to available aggregate data, including cancer type, adverse event severity, follow-up duration and treatment comparison. Median or mean follow-up duration was extracted where available and explored as a covariate in meta-regression to evaluate whether longer observation periods (≥3 years) influenced survival estimates. Given the absence of individual patient data, effect estimates remained unadjusted for patient-level confounders.

The GRADE system was used to evaluate evidence quality. We assessed the studies by analyzing how the authors reported the treatment population, the control population, the reporting of endocrine adverse-effects’ incidence in both groups, and the potential for selection and information bias (Appendix A). All analyses were performed using R statistical computing software (version 2024.12.1+563.pro5). A *p*-value < 0.05 was considered statistically significant.

## 3. Results

The systematic search identified 43 studies (10 prospective, 32 retrospective, and 1 with both retrospective and prospective components), encompassing a total of 17,399 patients. Study characteristics are summarized in Appendix A.

### 3.1. Stratification by ICI Class (PD-1/PD-L1 vs. CTLA-4)

Across all studies, thyroid dysfunctions were the most frequently reported endocrine irAEs, occurring in approximately 35–45% of patients treated with PD-1/PD-L1 inhibitors, compared with 5–15% under CTLA-4 blockade. Hypophysitis was strongly associated with CTLA-4 inhibitors, reported in 10–68% of treated patients vs. 2–8% in the PD-1/PD-L1 cohorts. Adrenal insufficiency was less common overall (4–14%), but more often observed under anti-CTLA-4 therapy.

When stratified by tumor type, the pattern remained consistent. In melanoma, PD-1 inhibitors (nivolumab, pembrolizumab) were mainly linked to thyroid dysfunction (40–45%) and autoimmune diabetes (10–15%), while anti-CTLA-4 agents (ipilimumab, tremelimumab) were predominantly associated with hypophysitis (up to 60%) and adrenal insufficiency (6–11%). In NSCLC, PD-L1 inhibitors (atezolizumab, durvalumab) were primarily associated with thyroiditis or hypophysitis (10–16%), whereas anti-PD-1 agents showed higher rates of CIADM (11–13%), adrenal insufficiency (10–12%), and thyroid dysfunction (30–38%). In urothelial carcinoma, the frequency of thyroiditis was similar between anti-PD-1 and -PD-L1 therapies (10–15%), although hypothyroidism appeared somewhat more frequent with PD-L1 blockade (up to 13%).

### 3.2. Stratification by Adverse Events and Therapeutic Agent

Across studies, thyroid dysfunction was consistently the most frequently reported endocrine irAE, while hypophysitis and adrenal insufficiency were less common. Exact prevalence varied substantially between studies, reflecting differences in patient selection, monitoring frequency, and drug combinations. Therefore, the frequency data are presented descriptively rather than as pooled prevalence estimates. Most reported cases of adrenal insufficiency were secondary to pituitary dysfunction rather than primary adrenalitis. Imaging evidence of pituitary inflammation was infrequently available, reflecting the limited use of MRI or ACTH stimulation testing in retrospective studies. Consequently, the term “hypophysitis” in the literature often encompasses both confirmed inflammatory and functional pituitary disorders.

Further analyses examined the frequencies of endocrine-related dysfunctions per ICI, namely anti-CTLA-4 (ipilimumab, tremelimumab), anti-PD-L1 (atezolizumab, durvalumab), and anti-PD-1 (pembrolizumab, nivolumab).

When stratified by therapeutic agent, distinct patterns of endocrine adverse events became evident. PD-1 inhibitors such as pembrolizumab and nivolumab, were most frequently associated with thyroiditis (41–46%) and hypothyroidism (37–38%); hyperthyroidism was also observed (29–32%). Hypophysitis occurred less often (6–8%), while type 1 diabetes (11–13%), type 2 diabetes (17–18%), autoimmune pancreatitis (up to 21%), and adrenal insufficiency (30–35%) were reported at variable frequencies. These events typically emerged within the first 5–12 weeks for thyroid disorders, whereas diabetes and adrenal insufficiency tended to occur later, sometimes beyond 24 weeks.

PD-L1 inhibitors, including atezolizumab and durvalumab, showed a different profile: thyroiditis was not observed, while hypothyroidism reached 13% and hypophysitis 16%. Type 1 diabetes was occasionally reported (up to 39%), whereas type 2 diabetes was not observed. Autoimmune pancreatitis appeared in up to 29% of cases, and adrenal insufficiency ranged from 4 to 14%. The onset of these events was more variable, with some (such as hypophysitis) arising later in treatment.

CTLA-4 inhibitors (ipilimumab, tremelimumab) were characterized by a high frequency of hypophysitis, reported in 25–38% of cases, often appearing later (15–20 weeks after treatment initiation). Thyroid disorders were less common, with thyroiditis reported at 1–2%, hypothyroidism at up to 11%, and hyperthyroidism at 13–20%. Type 1 diabetes occurred in up to 17% and type 2 diabetes in up to 25%, while autoimmune pancreatitis (37–43%) and adrenal insufficiency (6–11%) were also described.

### 3.3. Stratification by Adverse-Event Onset

Reporting of onset timing was inconsistent across studies; therefore, pooled time-to-event analyses were not feasible. However, available data indicated a reproducible temporal sequence in endocrine irAE development. Thyroiditis typically appeared earliest, with median onset between 5 and 8 weeks after ICI initiation, followed by hyperthyroidism (8–10 weeks) and hypothyroidism (10–14 weeks). Adrenal insufficiency and hypophysitis were reported later in treatment, with median onset around 15–21 weeks, whereas immune-mediated diabetes mellitus (CIDM) showed the latest presentation, often after 24 weeks and in some cases beyond 70 weeks of exposure.

By tumor type, similar trends were observed: in melanoma, thyroid dysfunctions developed within 5–6 weeks, hypophysitis around 10 weeks, and adrenal insufficiency near 15 weeks; in NSCLC, most endocrine irAEs clustered around 10–15 weeks, while in urothelial carcinoma, thyroid disorders and adrenal insufficiency emerged around 11–12 weeks, and diabetes could appear much later.

Despite inter-study variability, these findings suggest a broadly consistent onset hierarchy—early thyroid dysfunction, intermediate hypophysitis and adrenal insufficiency, and late-onset diabetes—with low-to-moderate certainty due to heterogeneous reporting and variable endocrine monitoring schedules.

### 3.4. Correlation of Endocrine irAEs with ICI Response and Survival

Patients under ICI treatment demonstrated improved outcomes (OS and PFS) across melanoma, NSCLC, and urothelial cancer subtypes (Figure 2 and Figure 3). In the pooled data from the 43 studies analyzed herein, patients receiving treatment with ICIs had a 40% reduced risk of death (HR: 0.60; 95% CI: 0.54–0.67; *p* < 0.01; Figure 4) and improved PFS (HR: 0.61; 95% CI: 0.54–0.68; *p* < 0.01; Figure 5) compared with patients without ICI treatment. Between-study heterogeneity was moderate (I^2^ = 67.9% in Figure 4, I^2^ = 76.9% in Figure 5), reflecting expected clinical variability across tumor types and treatment regimens. Anti-CTLA-4 agents in melanoma significantly improved disease control (OR: 1.55; 95% CI: 1.37–1.77; *p* < 0.0001; Figure 6). Sensitivity analyses excluding high-risk studies did not materially alter the pooled estimates, confirming the robustness of the findings.

Residual heterogeneity variance was examined via funnel plots for each pooled analyses (Appendix A) and for the overall analysis (Appendix A). Egger’s test was statistically significant for asymmetry for OS (*p* = 0.008), but not for PFS (*p* = 0.169) coherent with funnel plot examination.

Pooled OR analysis revealed that anti-PD-L1 agents in urothelial cancer had a higher risk of irAEs (OR: 1.13; 95% CI: 0.86–1.49; *p* = 0.41), while pembrolizumab in NSCLC caused fewer irAEs than nivolumab (OR: 0.32; 95% CI: 0.32–1.02; *p* = 0.09).

Further subgroup analysis linked grade 3–4 endocrine irAEs with anti-PD-L1 agents in urothelial cancer (OR: 6.23; 95% CI: 2.98–13.1; *p* < 0.01), as well as with pembrolizumab in NSCLC (OR: 2.78; 95% CI: 0.97–7.31; *p* = 0.07) and melanoma (OR: 2.06; 95% CI: 1.18–3.73; *p* = 0.015; Appendix A).

In the univariate meta-regression model including all 43 studies, HR for PFS was not significantly associated with OR for adverse events (estimate = 0.20, 95% CI: −0.10 to 0.51; *p* = 0.193). This model explained less than 1% of between-study heterogeneity (R^2^ = 0.64%), with substantial residual heterogeneity (τ^2^ = 0.53; I^2^ = 96.7%; Appendix A).

In contrast, the multivariate meta-regression model, which incorporated cancer type, adverse event severity, and treatment comparison as covariates, significantly improved the model fit and accounted for a considerable proportion of heterogeneity, though without substantial asymmetry (R^2^ = 55.9%, τ^2^ = 0.23, I^2^ = 92.7%; Appendix A). In this model, NSCLC studies showed significantly lower odds of irAEs compared to melanoma (estimate = −0.60, *p* < 0.0001), while urothelial studies had higher odds (estimate = 0.66, *p* = 0.0049). In further univariate analysis, Grade 3–4 irAEs were strongly associated with higher probability of occurrence in patients receiving pembrolizumab (estimate = 0.88, *p* < 0.0001), vs. those receiving nivolumab (estimate = 0.43, *p* = 0.0023), comparisons that were significantly predictive for irAEs incidence.

Similarly, the association with OS was weak and not statistically significant (estimate = 0.14, *p* > 0.05). Multivariate analysis, however, explained 56% of heterogeneity (R^2^ = 55.9%) and confirmed cancer type, irAE severity, and ICI class as significant moderators, whereas the direct effect of irAE incidence remained non-significant.

These findings are visualized in bubble plots of OR for endocrine irAEs against HR for PFS and OS, respectively (Figure 7). No linear association was observed, suggesting that the occurrence of irAEs and clinical efficacy may be governed by partially independent mechanisms.

Risk of bias was assessed using the Cochrane RoB 2.0 tool for randomized trials and ROBINS-I for observational studies. Of the 43 included studies, 24 were rated as low, 15 as moderate, and 4 as high risk of bias. The main concerns involved blinding and incomplete outcome data in retrospective cohorts. Excluding high-risk studies did not materially affect pooled estimates, indicating overall methodological robustness (Appendix A).

The certainty of evidence, evaluated through the GRADE framework, was moderate for OS and PFS, supported by consistent findings across cancer types and study designs. Certainty was lower (low to moderate) for specific endocrine irAEs due to heterogeneity and imprecision in rare events. Overall, the evidence provides moderate confidence in the observed associations between endocrine irAEs and improved clinical outcomes (Appendix A).

## 4. Discussion

This systematic review and meta-analysis represents a comprehensive study to date of the relationship between endocrine irAEs and clinical outcomes in patients with melanoma, NSCLC, and urothelial cancer treated with ICIs. By analyzing data from 43 cohorts trials, comprising *n* = 17,399 patients, we aim to demonstrate that the occurrence of endocrine irAEs can be related to improved PFS and OS, suggesting their potential use as additional surrogate biomarkers of immunotherapy efficacy.

Endocrine irAEs associated with anti-PD-1 therapies predominantly included thyroid disorders, while hypophysitis was more frequently linked to anti-CTLA-4 agents. Autoimmune diabetes and adrenal insufficiency were less common, but more often observed with anti-PD-1 or anti-PD-L1 therapies. Thyroid dysfunction, thyroiditis, and elevated anti-thyroid antibodies showed a strong association with improved OS and PFS, making these hormonal alterations potential markers of a robust immune response. The onset of irAEs varied by treatment, occurring approximately 7 weeks after ipilimumab and 10–15 weeks after nivolumab initiation. The timing and presentation of endocrine disorders reflect the distinct pharmacological actions of each ICI. Partial recovery of the gonadal and thyroid-pituitary axis function has been observed in 50–60% of cases, though adrenal axis recovery remains limited [66].

The pathophysiology of endocrine irAEs reflects immune hyperactivation against self-antigens, the induction of self-reactive antibodies, and cytokine dysregulation. Anti–CTLA-4 antibodies mobilize naïve T cells, inducing lymphocytic infiltration of the pituitary gland and leading to hypophysitis and secondary adrenal insufficiency. This mechanism resembles type II and IV hypersensitivity, mediated by cytotoxic T lymphocytes and antibody-dependent complement activation. In contrast, PD-1 and PD-L1 inhibitors primarily enhance peripheral effector T-cell function, predisposing to thyroiditis and, rarely, autoimmune diabetes. Elevated anti-thyroid peroxidase and anti-thyroglobulin antibodies have been observed in up to 40% of patients prior to overt thyroid dysfunction, suggesting B-cell participation in antigen cross-reactivity [67]. The delayed onset of diabetes mellitus and adrenalitis, often occurring after several months of therapy, indicates a cumulative autoimmune process rather than acute cytotoxic injury. Despite these mechanistic distinctions, endocrine irAEs share the unifying feature of irreversible glandular destruction, which contrasts with the often transient nature of dermatologic or gastrointestinal irAEs. Ferrara et al. [68] underscore that the association of endocrine irAEs with improved clinical outcomes reflects the activation of robust immune responses; yet, as reported by Fukushima et al. [69] further investigation is needed to confirm whether irAEs reliably predict ICI efficacy.

Our subgroup analyses reinforce that ICI treatment is linked to improved OS and PFS across melanoma, NSCLC, and urothelial cancer. Patients recieving ICI treatment demonstrated a 41% improvement in survival outcomes compared to those without irAEs. However, regarding the predictive value of endocrine irAEs, this may be influenced by several factors, such as the incidence of relatively rare endocrine irAEs, like adrenal insufficiency and autoimmune diabetes in ICI-treated patients, which may limit their prognostic utility. Geographical scope should also be underscored, as the included studies were conducted in North America, Europe, and Asia, ensuring global applicability for melanoma and NSCLC, but providing limited data for urothelial cancer. However, despite differences in study design and population characteristics and the moderate heterogeneity documented (Heterogeneity: I^2^ = 97.0% for PFS and I^2^ = 84.6% for OS), the overall trends support an eventual link between endocrine irAEs and ICI-induced immune activation.

Our meta-regression analysis did not reveal a significant association between clinical efficacy (as measured by PFS HRs) and the occurrence of endocrine irAEs (OR), either in univariate or multivariate settings. This suggests that treatment benefit and toxicity profiles may operate independently across ICI trials, aligning with prior findings that point to distinct biological underpinnings [7]. Notably, the multivariate model identified several important moderators of irAE risk. Cancer type was strongly associated with adverse event profiles, with NSCLC studies exhibiting significantly lower odds and urothelial studies higher odds compared to melanoma. Furthermore, grade 3–4 adverse events were substantially more likely than lower-grade events, and pembrolizumab-based regimens were associated with increased irAE odds compared to nivolumab. These findings emphasize the imperative need for stratified safety monitoring based on cancer type and drug class, and reinforce the concept that efficacy and toxicity should be evaluated independently when selecting ICI regimens.

We further conducted unadjusted (univariate) meta-regression analyses due to limitations in available study-level covariates. While this approach provides exploratory insights into the relationship between efficacy and toxicity, results may be subject to confounding by unmeasured study characteristics such as disease stage, prior therapy, or comparator type. Future analyses incorporating more granular covariate data may help clarify these associations.

Severe irAEs may necessitate treatment discontinuation or lead to significant morbidity, emphasizing the importance of early detection and tailored management strategies to mitigate negative outcomes while maximizing therapeutic benefits. The complexity of endocrine irAEs is compounded by inconsistent terminology in clinical trials. Terms like “hypopituitarism,” “hypophysitis,” and “adrenal insufficiency” are often used interchangeably or ambiguously. For example, decreased ACTH levels may indicate adrenal insufficiency, adrenalitis, or hypophysitis, depending on the context [13,14]. Similarly, cases labeled as “transient thyroiditis” or “sick euthyroid syndrome” may represent expected immune responses rather than pathologic events [16,70]. Additionally, studies often fail to provide clear criteria for defining and resolving endocrine AEs. The assumption that resolution equates to glandular recovery is problematic, as many endocrine irAEs involve permanent dysfunction requiring lifelong hormone replacement [71].

Estimating the true frequency of endocrine irAEs is inherently challenging. Many studies included systematic thyroid monitoring but lacked comprehensive pituitary or adrenal testing, leading to under-detection of subclinical events. Differentiating between hypophysitis, hypopituitarism, and primary adrenal insufficiency remains challenging in clinical practice and across published cohorts. Anti-CTLA-4 agents induce lymphocytic infiltration of the pituitary leading to secondary adrenal and thyroid failure, whereas primary immune adrenalitis—characterized by isolated cortisol deficiency with preserved pituitary function—is exceedingly uncommon. This diagnostic overlap further complicates reporting, as imaging confirmation of pituitary inflammation is rarely available [14,20]. Furthermore, many studies lacked systematic pituitary hormone panels or imaging, relying instead on symptomatic presentation, which underestimates subclinical cases. Dosage and regimen heterogeneity also influence toxicity: higher-dose ipilimumab (10 mg/kg) and combination nivolumab–ipilimumab regimens show greater endocrine toxicity than monotherapy. Consequently, our reported ranges should be viewed as qualitative patterns rather than exact incidence figures.

Previous systematic reviews have explored the relationship between irAEs and survival outcomes, but they focused on specific cancer types or therapies. For example, Zhao et al. evaluated 26 NSCLC studies, while Johnson et al. analyzed 11 melanoma trials [72,73]. Our inclusion of 43 studies from 3 cancer types where ICIs are used as first-line therapy, enhances the robustness of our findings, providing a more detailed understanding of endocrine irAEs in melanoma, NSCLC, and urothelial cancer.

Three key studies investigating endocrine irAEs in cancer patients treated with ICIs provide critical insights that support and contextualize the findings of our meta-analysis. Zhang et al. conducted a multicenter retrospective study in 380 NSCLC patients treated with pembrolizumab and reported endocrine irAEs in 30% of patients, the majority being thyroid dysfunction (28.2%), with subclinical hyperthyroidism (44.9%) and subclinical hypothyroidism (34.6%) as the most common phenotypes [37]. Hypophysitis and type I diabetes were less frequent (2.4% and 0.8%, respectively), and the presence of irAEs—particularly subclinical hypothyroidism—was independently associated with prolonged PFS. Oppolzer et al. retrospectively analyzed 102 patients with metastatic urothelial or renal cell carcinoma treated with ICIs and found a total endocrine irAE incidence of 18.6%, predominantly thyroiditis (13.7%) and hypophysitis (3.9%), with the latter exclusively observed in patients receiving dual ICI therapy. Importantly, 73% of endocrine irAEs required permanent hormone replacement [74].

Similarly, Kassi et al. conducted a prospective study of 339 metastatic melanoma patients and found an overall endocrine irAE incidence of 11.8%, with isolated hypophysitis in 6.2% and thyroid dysfunction in 5.6%. Combination or sequential therapy was associated with a significantly higher incidence (18.5%) compared to anti-CTLA-4 (5%) or anti-PD-1/PD-L1 (13.4%) monotherapy [75]. These findings are in agreement with our pooled data showing increased endocrine toxicity with anti-PD-L1 agents in urothelial cancer (OR: 6.23) and pembrolizumab in NSCLC (OR: 2.78), but also improved survival outcomes in patients experiencing endocrine irAEs (HR for OS: 0.60; HR for PFS: 0.61). Across all three studies and our meta-analysis, the occurrence of endocrine irAEs—especially thyroid dysfunction and hypophysitis—emerged as a clinically significant marker of immune activation and favorable prognosis, underscoring the need for routine endocrine monitoring and interdisciplinary management in patients undergoing ICI therapy.

Contradictory findings in the literature may arise from methodological heterogeneity. Some studies included non-systematic endocrine monitoring, leading to under-reporting of subclinical cases, while others pooled CTLA-4 and PD-1 inhibitors despite distinct toxicity profiles. Cheung et al. reported that overt thyroid dysfunction during anti–PD-1 therapy was independently associated with longer PFS and OS in pooled analyses of more than 5000 patients with solid tumors [76]. Similarly, Indini et al. observed that melanoma patients experiencing irAEs under nivolumab or pembrolizumab had significantly improved outcomes compared with those without such events [77]. Moreover, the timing of endocrine evaluation varied widely, from baseline-only screening to monthly follow-up, complicating cross-trial comparability. These discrepancies emphasize the need for standardized definitions and monitoring schedules in future ICI studies.

Our meta-regression results further suggest that, although endocrine irAEs coincide with favorable survival, they do not directly predict treatment efficacy when cancer type, drug class, and event severity are considered. This observation aligns with recent large-scale pooled analyses showing that irAEs reflect systemic immune activation rather than a strict biomarker of therapeutic success. A plausible hypothesis—supported by prior literature—is that endocrine irAEs could, in some settings, represent a surrogate of heightened antitumor immunity (e.g., immune activation signatures observed with ICIs), yet this relationship may be context-dependent and modified by tumor histology, regimen, and timing of irAE onset. Future work should test this prospectively, using standardized endocrine panels, longitudinal biomarker assessment, and time-dependent modeling of time-to-irAE and time-to-event outcomes to minimize immortal-time and detection biases [38].

Several limitations should be acknowledged. First, the analysis was conducted at the study level using aggregated data, precluding adjustment for patient-level confounders such as disease stage, performance status, or prior therapy. Second, substantial heterogeneity was present in endocrine monitoring, event reporting, and follow-up duration across studies. Third, the lack of uniform criteria distinguishing primary from secondary adrenal insufficiency or transient from chronic thyroid dysfunction may have led to misclassification. Fourth, publication bias may have influenced the results due to heterogeneity among the included studies; however, sensitivity analyses suggest that these effects are minimal (I^2^ < 80% in all statistical analyses and in some pooled analyses even <50%). Residual heterogeneity likely reflects inconsistent endocrine screening, non-standardized dosing regimens, and variable case definitions (e.g., “hypophysitis” vs. secondary adrenal insufficiency), which limit cross-trial comparability. Lastly, the predominance of retrospective designs in NSCLC and melanoma, compared to the largely prospective nature of urothelial cancer studies, reflects the experimental phase of ICIs in urothelial carcinoma and may introduce inherent biases and consequently limit causal inference. Although long-term survival (≥3 years) is a critical benchmark for durable ICI responses, not all studies reported sufficient follow-up to permit stratified analyses. Our findings, however, remained consistent across studies with both shorter and longer median follow-up, suggesting that the association between endocrine irAEs and improved PFS/OS is robust but should be interpreted within the constraints of variable follow-up durations. Despite these limitations, the overall consistency of findings across tumor types and treatment regimens supports the robustness of the observed associations.

While pooled analyses showed better OS and PFS among patients who received ICIs, these contrasts do not establish causality with the development of endocrine irAEs. In our study, meta-regression did not identify a significant association between irAE incidence and efficacy after accounting for cancer type, ICI class, and event severity, indicating that toxicity and benefit may be only partially coupled at the study level. This should temper causal interpretations drawn from subgroup or unadjusted comparisons.

In our dataset analysis, despite favorable pooled survival contrasts, the meta-regression was null for a direct irAE–efficacy link; thus, our findings argue for cautious interpretation and for designing studies that can disentangle correlation from causation (e.g., individual-patient data analyses with time-varying covariates, Mendelian randomization where feasible). Until such evidence is available, endocrine irAEs should be viewed as hypothesis-generating markers as a parallel manifestation of immune activation and not definitive surrogates of benefit.

## 5. Conclusions

ICIs have transformed the treatment of solid tumors but frequently trigger endocrine irAEs. Although these toxicities are often persistent, timely recognition and appropriate management—through monitoring, diagnostic testing, and hormone replacement—generally permit continuation of therapy with limited impact on patients’ quality of life. In this review, 11–30% of patients developed endocrinopathies, with the highest incidence and severity observed under combined or sequential anti-CTLA-4 and anti-PD-1/PD-L1 regimens. As the clinical use of ICI use expands, the burden of endocrine irAEs will increase, underscoring the need for intensified surveillance, closer collaboration with endocrinologists, and research into underlying mechanisms, patient-specific risk factors, and improved preventive strategies.

## Figures and Tables

**Figure 1 cancers-17-03675-f001:**
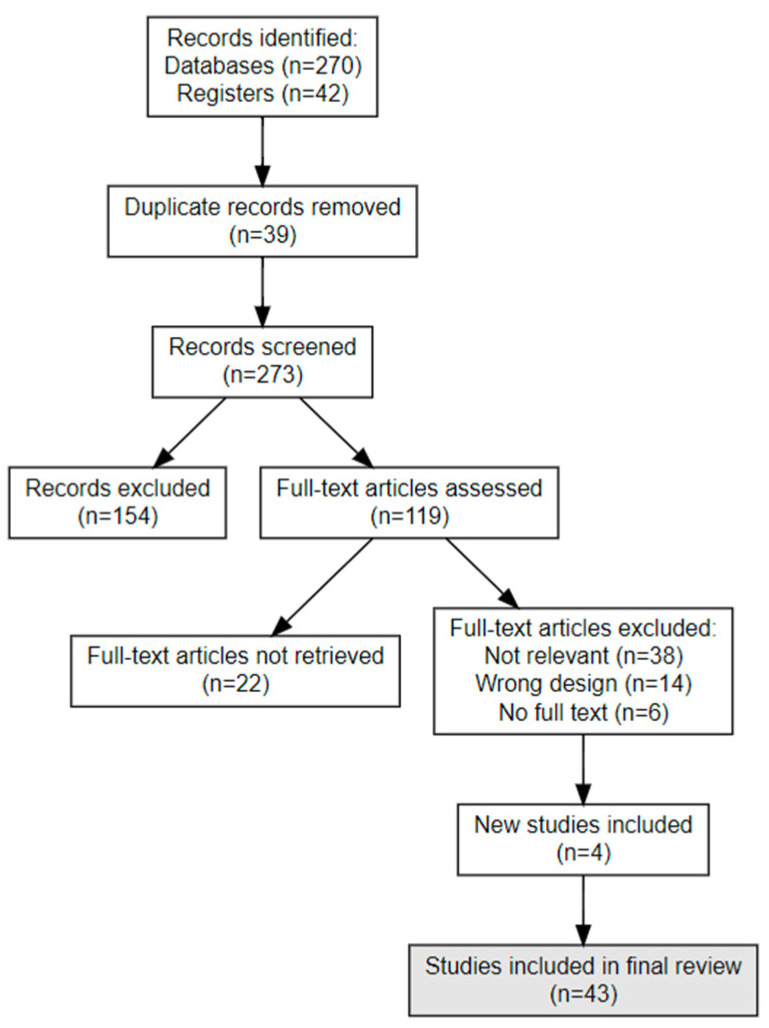
PRISMA Flow Chart—Search Strategy (Sum-up diagram).

**Figure 2 cancers-17-03675-f002:**
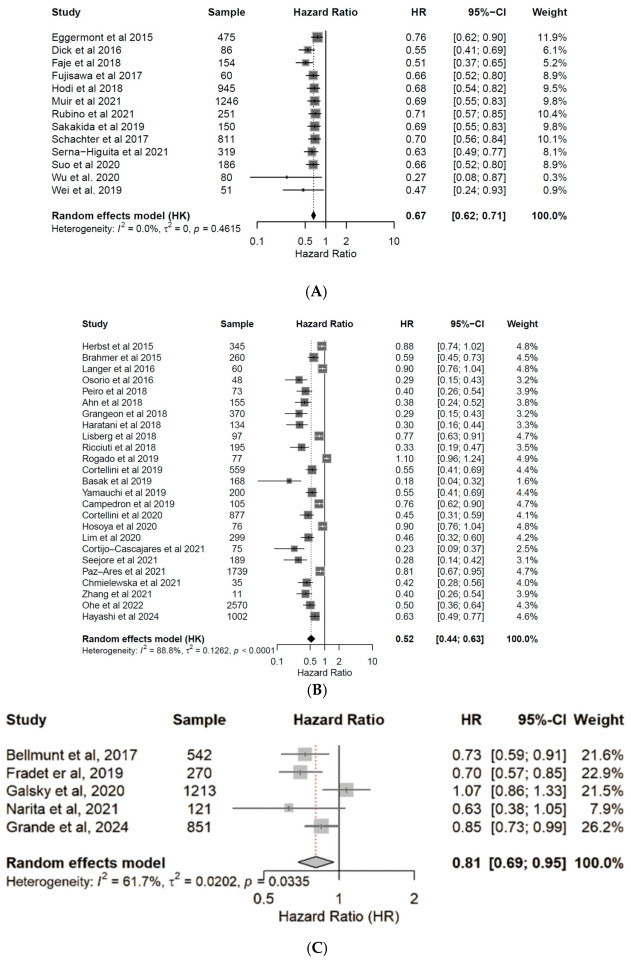
Forest plots for overall survival (OS). Pooled hazard ratios of OS in patients with melanoma [24,25,26,27,28,29,30,31,32,33,34,35,36] (**A**), NSCLC [37,38,39,40,41,42,43,44,45,46,47,48,49,50,51,52,53,54,55,56,57,58,59,60,61] (**B**), and urothelial cancer [50,62,63,64,65] (**C**), treated with anti-PD-(L)1 or CTLA-4 antibodies. CI, confidence interval; HR, hazard ratio.

**Figure 3 cancers-17-03675-f003:**
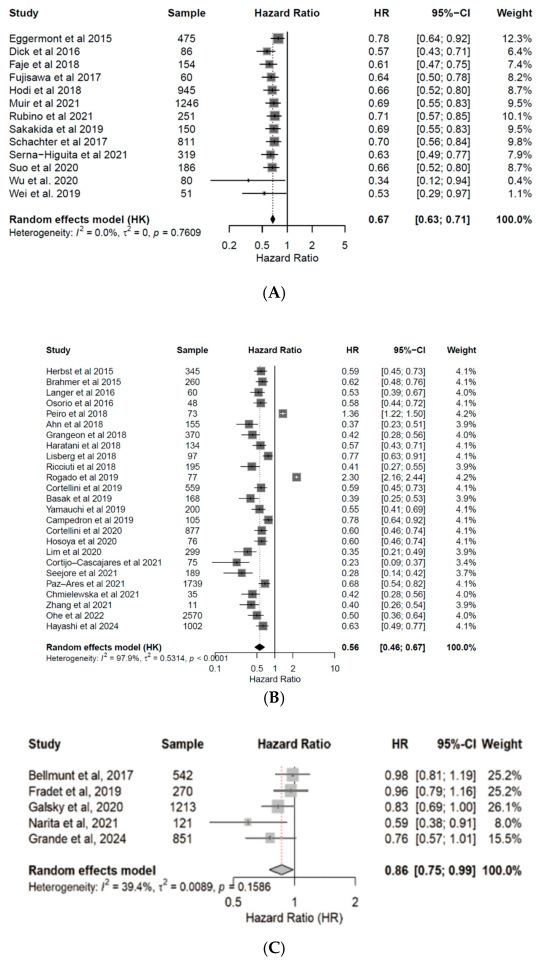
Forest plots for progression free survival (PFS). Pooled hazard ratios of PFS in patients with melanoma [24,25,26,27,28,29,30,31,32,33,34,35,36] (**A**), NSCLC [37,38,39,40,41,42,43,44,45,46,47,48,49,50,51,52,53,54,55,56,57,58,59,60,61] (**B**), and urothelial cancer [50,62,63,64,65] (**C**), treated with anti-PD(L)-1 or CTLA-4 antibodies. CI, confidence interval; HR, hazard ratio.

**Figure 4 cancers-17-03675-f004:**
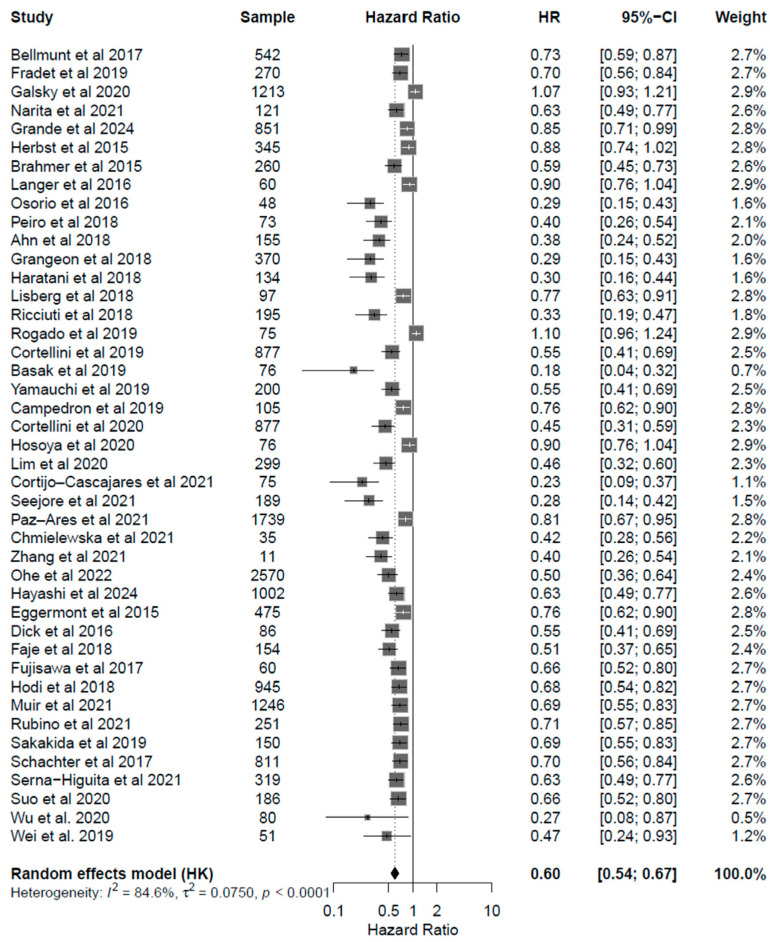
Forest plot for overall survival (OS) including all studies analyzed [24,25,26,27,28,29,30,31,32,33,34,35,36,37,38,39,40,41,42,43,44,45,46,47,48,49,50,51,52,53,54,55,56,57,58,59,60,61,62,63,64,65]. The pooled HR was 0.60 (95% CI: 0.54–0.67; *p* < 0.001), indicating a 40% reduction in the risk of death. Moderate heterogeneity was observed (I^2^ = 84.6%).

**Figure 5 cancers-17-03675-f005:**
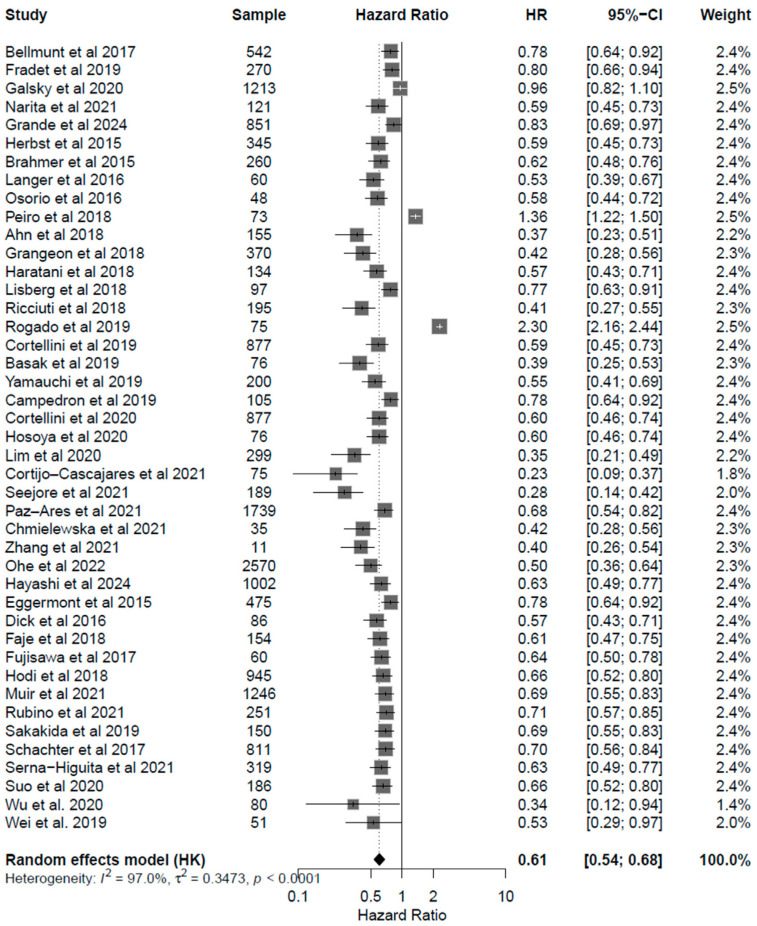
Forest plot for progression free survival (PFS) including all studies analyzed Mendeley, showing pooled HRs for PFS associated with endocrine irAEs (HR: 0.61, 95% CI: 0.54–0.68; *p* < 0.001). Results demonstrate consistent PFS benefit across melanoma, NSCLC, and urothelial carcinoma cohorts, supporting the hypothesis that endocrine irAEs coincide with a more active immune response and enhanced tumor control.

**Figure 6 cancers-17-03675-f006:**
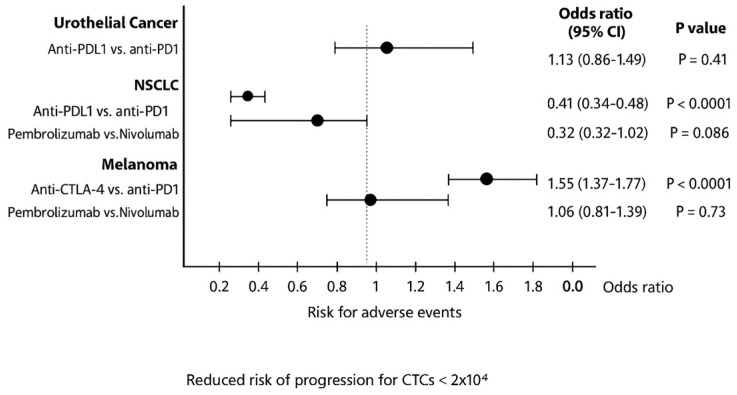
Pooled odds ratios of risk for endocrine adverse events in patients with urothelial cancer, NSCLC and melanoma treated with different ICIs. CI, confidence interval.

**Figure 7 cancers-17-03675-f007:**
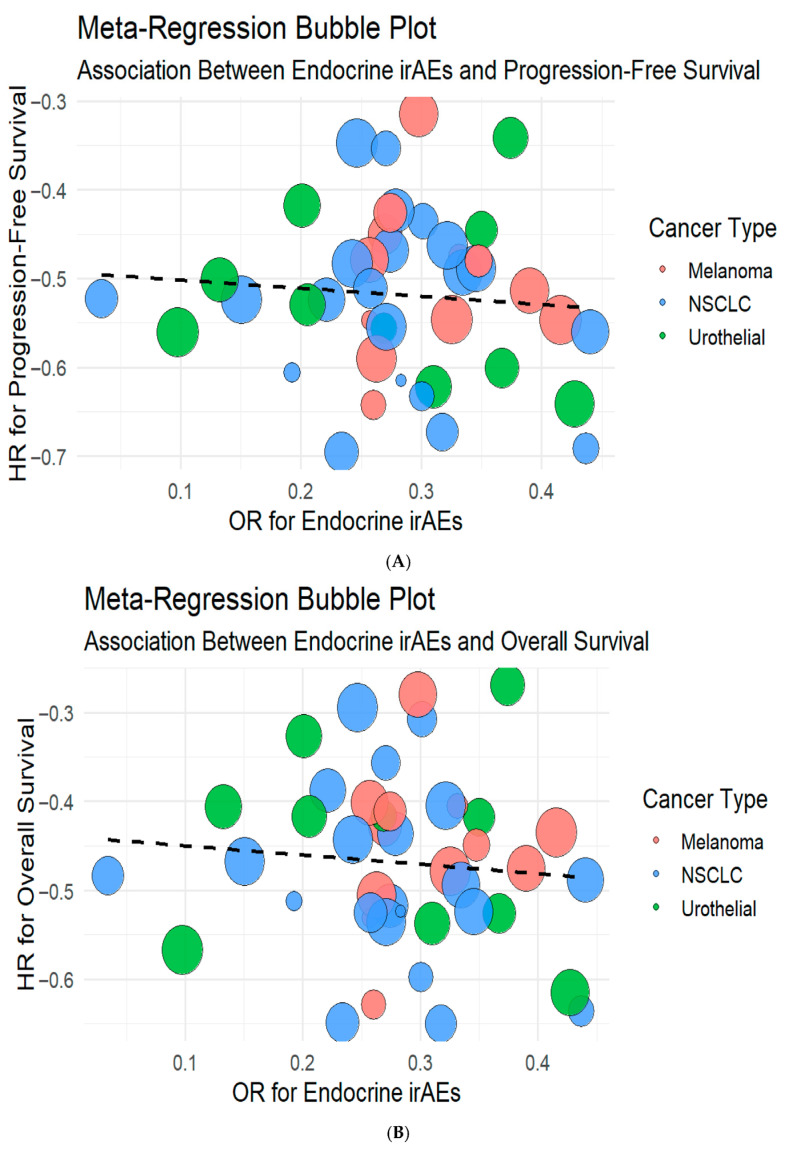
Meta-regression bubble plots illustrating the association between endocrine irAEs with progression-free survival (**A**) and overall survival (**B**). The size of the bubbles represents study weight, while color coding indicates cancer type. No significant linear correlation was observed between endocrine irAE incidence and survival benefit (PFS estimate = 0.20, 95% CI −0.10 to 0.51, *p* = 0.19; OS estimate = 0.14, *p* > 0.05), suggesting independent mechanisms underlying efficacy and toxicity.

**Table 1 cancers-17-03675-t001:** PICO statement.

P(opulation)	Patients with melanoma, NSCLC and urothelial cancer
I(ntervention)	Measurement of endocrine adverse events during patients’ follow-up
C(omparator)	Immune checkpoint inhibitors (anti-PD-(L)1 or anti-CTLA-4 agents)
O(utcome)	Progression Free Survival (PFS), Overall Survival (OS)

## Data Availability

All data supporting the findings of this systematic review and meta-analysis are available within the article and the Appendix A. Extracted study-level data used for quantitative synthesis were obtained from previously published articles included in this review. No new patient-level data were generated. The R scripts used for the statistical analyses and meta-analytic computations (including forest plots, funnel plots, heterogeneity models, and meta-regression) are available from the corresponding author upon reasonable request and will be uploaded to a public repository following manuscript acceptance.

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
