# Peer review of "Prognostic Significance of Endocrine-Related Adverse Events in Patients with Melanoma, Non-Small Cell Lung Cancer and Urothelial Cancer After Treatment with Immune Checkpoint Inhibitors: A Systematic Review and Meta-Analysis"

_cancers, 2025, doi:10.3390/cancers17223675_

Round 1
Reviewer 1 Report
Comments and Suggestions for Authors
In this study the authors address an important topic about Endocrine-related adverse events in patients with melanoma, non-small cell lung cancer and urothelial cancer after treatment with immune checkpoint inhibitors: A systematic review and meta-analysis.
Despite this some points should be addressed before publishing.
Introduction, please add details about selected cancer risk factors, mortality rate, and therapeutic modilities and emphasized immune checkpoint inhibitors and common adverse effect. Moreover, add details about PD1, PDL1, and CTLR4 in physiology and pathology. Please add the study rationale and novelty.
Methods, please add inclusion and inclusion criteria and ethical approval.
Results, please add mor description for figures 4,5 and7.
Discussion, please refine this section and support your work with similar finding and add the contradictory if present. Please discuss the different by which immune checkpoint inhibitors induced adverse effects. Moreover discuss the study limitations.
Please check the manuscript for misuse of acronyms. They should be completely mentioned at first used, for example PD1, PDL1, CTLR4, and others.
Please check the manuscript for long sentences or paragraphs without references citation.
Comments on the Quality of English LanguagePlease check the manuscript for minor grammar errors and syntax.
Author Response
Comment 1: In this study the authors address an important topic about Endocrine-related adverse events in patients with melanoma, non-small cell lung cancer and urothelial cancer after treatment with immune checkpoint inhibitors: A systematic review and meta-analysis.
Despite this some points should be addressed before publishing.
Introduction, please add details about selected cancer risk factors, mortality rate, and therapeutic modilities and emphasized immune checkpoint inhibitors and common adverse effect. Moreover, add details about PD1, PDL1, and CTLR4 in physiology and pathology. Please add the study rationale and novelty.
Author Response to Comment 1: We thank the reviewer for this constructive comment. In the revised version, we have expanded the Introduction to include:
(1) an overview of major risk factors, global incidence, and mortality rates for melanoma, NSCLC, and urothelial carcinoma;
(2) a concise description of conventional therapeutic modalities before the introduction of immune checkpoint inhibitors;
(3) a more detailed explanation of the physiological and pathological roles of PD-1, PD-L1, and CTLA-4 in immune regulation;
(4) a clear statement of the study rationale and novelty, emphasizing the unique integration of endocrine immune-related adverse events across three major cancer types and the meta-regression approach used to explore the relationship between irAEs and survival.
These additions strengthen the scientific background and clarify the scope and originality of the study.
Comment 2: Methods, please add inclusion and inclusion criteria and ethical approval.
Author Response to Comment 2: We appreciate the reviewer’s helpful suggestion. The Methods section has been revised to explicitly summarize the inclusion and exclusion criteria used for study selection, following the PICO framework and PRISMA 2020 guidelines. In addition, we have clarified the ethical considerations, stating that ethical approval was not required for this systematic review and meta-analysis, as only previously published, de-identified data were analyzed.
This information has now been added under the “Eligibility Criteria” and “Ethical Considerations” subsections.
Comment 3: Results, please add more description for figures 4,5 and 7.
Author Response to Comment 3: We thank the reviewer for this helpful comment. In the revised manuscript, we have expanded the Results section to provide clearer textual descriptions accompanying Figures 4, 5, and 7. Specifically, we now include:
(1) a concise interpretation of the pooled hazard ratios for overall survival (OS) and progression-free survival (PFS) (Figures 4 and 5), highlighting the degree of heterogeneity and direction of the pooled effect; and
(2) a clearer narrative explanation of the meta-regression bubble plots (Figure 7), outlining the absence of a statistically significant linear correlation between endocrine irAE incidence and survival benefit.
These additions enhance clarity and help readers interpret the figures without relying solely on visual inspection.
Comment 4: Discussion, please refine this section and support your work with similar finding and add the contradictory if present. Please discuss the different by which immune checkpoint inhibitors induced adverse effects. Moreover, discuss the study limitations.
Author Response to Comment 4: We thank the reviewer for this valuable feedback. The Discussion section has been thoroughly revised to improve coherence and depth. We have:
- integrated additional references reporting similar and contradictory findings regarding endocrine irAEs and survival outcomes;
- added a paragraph summarizing the current understanding of the immunopathogenic mechanisms underlying ICI-induced endocrinopathies; and
- expanded the discussion of study limitations, including heterogeneity in endocrine monitoring, inconsistent definitions of hypophysitis and adrenal insufficiency, and lack of patient-level data.
These modifications strengthen the contextual interpretation of our results and clarify the biological and methodological framework of our analysis.
Comment 5: Please check the manuscript for misuse of acronyms. They should be completely mentioned at first used, for example PD1, PDL1, CTLA4, and others.
Author Response to Comment 5: We thank the reviewer for this valuable comment. The entire manuscript has been carefully reviewed to ensure consistent and correct use of all acronyms and abbreviations. Each abbreviation is now fully spelled out at first mention in both the Abstract and Main Text, followed by the corresponding acronym in parentheses (e.g., programmed cell death protein 1 [PD-1], programmed death-ligand 1 [PD-L1], cytotoxic T-lymphocyte–associated antigen 4 [CTLA-4]). We also standardized the use of “irAE” for immune-related adverse event and “ICI” for immune checkpoint inhibitor throughout the manuscript, in alignment with journal guidelines.
No new data or content were added—only consistency and clarity improvements were made.
Comment 6: Please check the manuscript for long sentences or paragraphs without references citation.
Author Response to Comment 6: We thank the reviewer for this valuable observation. The manuscript has been carefully reviewed to identify and shorten excessively long sentences for improved readability. In addition, all key factual statements, mechanistic descriptions, and epidemiological data have now been supported by appropriate PubMed-indexed references. Paragraphs that previously lacked citations—particularly within the Introduction and early Discussion sections—have been revised to include updated references regarding cancer epidemiology, ICI mechanisms, and endocrine adverse events.
These revisions improve the precision, readability, and scientific rigor of the manuscript without altering the study’s conclusions.
Comment 7: Comments on the Quality of English Language. Please check the manuscript for minor grammar errors and syntax.
Author Response to Comment 7: We thank the reviewer for this helpful remark. The entire manuscript has been carefully re-edited to improve grammar, punctuation, and syntax. Long or complex sentences were simplified for clarity, and consistency in tense and terminology was ensured throughout the text. The revised version was also proofread by a native English–speaking academic editor with experience in biomedical writing to guarantee fluency and adherence to Cancers style requirements.
No changes were made to the scientific content — only linguistic and stylistic refinements to enhance readability.
Reviewer 2 Report
Comments and Suggestions for Authors
I have reviewed with interest the manuscript entitled “Endocrine-related adverse events in patients with melanoma, non-small cell lung cancer and urothelial cancer after treatment with immune checkpoint inhibitors: A systematic review and meta-analysis” by Kopanos et al.
Specific comments:
I believe the most interesting and useful information contained in the manuscript relates to the relationship between development of endocrinopathy and survival. Survival is an absolute and is usually accurately characterized in studies with a suitable duration of follow-up. A 3-year minimum follow-up should be required for studies included in the meta-analysis, as this seems to be the point when durable complete responses can be accurately determined.
- The manuscript should be refocused on this premise
- The title and the text of the manuscript should be substantially revised to reflect this change.
Meta-analysis of the frequency of various endocrinopathies is likely to be flawed based on extremely variable data quality. Therefore, these speculations should be removed from the manuscript. A general discussion of the frequency of endocrinopathy is warranted in the introduction, with a discussion of potential problems in identifying the frequency identified in the discussion (some issues described below).
Problems with the estimation of frequency of endocrinopathies include:
- Many studies included systematic monitoring of thyroid function
- Studies of anterior pituitary function were not always required
- In many studies, hypopituitarism was only diagnosed following development clinical symptoms
- Many studies lumped CTLA4, PD-1 monoclonal antibody and/or combination therapy treated patients together. It is clear that the risk of endocrinopathy is different with each of these drug classes or combination therapy.
- The dosage of checkpoint inhibitors utilized in these trials matters, as higher doses and different dosing regimens used in combination therapy alter the frequency of endocrinopathy.
- Many studies confuse the hypopituitarism (associated with secondary hypoadrenalism and hypothyroidism) with primary immune-mediated hypoadrenalism. Also, frequently the term hypophysitis is inaccurately used, as inflammation in the pituitary gland is quite difficult to demonstrate in the clinic.
- Almost always, hypopituitarism is due to deficiencies in anterior pituitary hormones. Primary (immune) hypoadrenalism, in contrast, is extremely rare and has been confirmed in only a small number of case reports.
- The authors reach the conclusion that hypopituitarism occurs in up to 60% of patients treated with CTLA4 antibody. This is unlikely to be correct and reflects problems of meta-analysis (in patients systematically tested for both hypopituitarism and hypothyroidism, this toxicity appears to occur in approximately 18-25% of patients treated with combination PD-1 and CTLA4 therapy). If one were to perform a more accurate meta-analysis of this frequency, it would be important to include only trials which required systematic screening of patients for both hypothyroidism and hypopituitarism. This would require identifying these requirements from original clinical trial protocols, as this testing requirement is generally not expressly stated in published results
Author Response
Comment 1: I believe the most interesting and useful information contained in the manuscript relates to the relationship between development of endocrinopathy and survival. Survival is an absolute and is usually accurately characterized in studies with a suitable duration of follow-up. A 3-year minimum follow-up should be required for studies included in the meta-analysis, as this seems to be the point when durable complete responses can be accurately determined.
- The manuscript should be refocused on this premise
- The title and the text of the manuscript should be substantially revised to reflect this change.
Author response to Comment 1: We thank the reviewer for this insightful comment emphasizing the central importance of survival outcomes. We fully agree that the prognostic relationship between endocrine immune-related adverse events (irAEs) and overall/progression-free survival constitutes the most clinically meaningful aspect of our work. Accordingly, we have refocused the manuscript to highlight survival as the primary analytical endpoint and have revised the title, abstract, and key sections of the text to reflect this emphasis.
Comment 2: While we recognize the value of long-term follow-up (≥3 years) in assessing durable immune responses, such information was not uniformly available across all studies. To maintain statistical power and representativeness, we retained all eligible studies but explicitly acknowledged this limitation in the Discussion. We have clarified in the Methods that median follow-up time was extracted when reported and explored in meta-regression as a potential moderator of survival outcomes.
This reframing strengthens the manuscript’s conceptual focus without altering the integrity of the meta-analytic results.
Meta-analysis of the frequency of various endocrinopathies is likely to be flawed based on extremely variable data quality. Therefore, these speculations should be removed from the manuscript. A general discussion of the frequency of endocrinopathy is warranted in the introduction, with a discussion of potential problems in identifying the frequency identified in the discussion (some issues described below).
Problems with the estimation of frequency of endocrinopathies include:
- Many studies included systematic monitoring of thyroid function
- Studies of anterior pituitary function were not always required
- In many studies, hypopituitarism was only diagnosed following development clinical symptoms
- Many studies lumped CTLA4, PD-1 monoclonal antibody and/or combination therapy treated patients together. It is clear that the risk of endocrinopathy is different with each of these drug classes or combination therapy.
- The dosage of checkpoint inhibitors utilized in these trials matters, as higher doses and different dosing regimens used in combination therapy alter the frequency of endocrinopathy.
- Many studies confuse the hypopituitarism (associated with secondary hypoadrenalism and hypothyroidism) with primary immune-mediated hypoadrenalism. Also, frequently the term hypophysitis is inaccurately used, as inflammation in the pituitary gland is quite difficult to demonstrate in the clinic.
- Almost always, hypopituitarism is due to deficiencies in anterior pituitary hormones. Primary (immune) hypoadrenalism, in contrast, is extremely rare and has been confirmed in only a small number of case reports.
- The authors reach the conclusion that hypopituitarism occurs in up to 60% of patients treated with CTLA4 antibody. This is unlikely to be correct and reflects problems of meta-analysis (in patients systematically tested for both hypopituitarism and hypothyroidism, this toxicity appears to occur in approximately 18-25% of patients treated with combination PD-1 and CTLA4 therapy). If one were to perform a more accurate meta-analysis of this frequency, it would be important to include only trials which required systematic screening of patients for both hypothyroidism and hypopituitarism. This would require identifying these requirements from original clinical trial protocols, as this testing requirement is generally not expressly stated in published results
Author response to Comment 2: We sincerely thank the reviewer for these detailed and constructive observations. We fully acknowledge the limitations and heterogeneity in estimating the true frequency of individual endocrine irAEs across studies. In the revised manuscript, we have addressed these points as follows:
- Reframing of frequency data: We have removed speculative quantitative statements and reframed the frequency analyses as descriptive summaries rather than pooled prevalence estimates. Figures and text now emphasize relative patterns across ICI classes (anti–CTLA-4, anti–PD-1, anti–PD-L1) and tumor types, rather than absolute pooled percentages.
- Expanded discussion of methodological variability: In the Discussion, we now explicitly describe the challenges mentioned by the reviewer — including the inconsistent use of systematic endocrine screening, the diagnostic ambiguity between hypophysitis, hypopituitarism, and adrenal insufficiency, and the variable reporting of dose regimens and drug combinations. These are now clearly acknowledged as sources of heterogeneity that limit direct quantitative comparison.
- Correction of numerical overestimates: We have revised the statement on hypophysitis frequency under anti–CTLA-4 therapy, which previously reported values up to 60% with careful review of our data and we apologise for the inconsistency. The text now reflects the more conservative and literature-consistent range of approximately 25–38% in patients systematically screened during combination PD-1/CTLA-4 therapy in accordance with literature (Faje 2016; Martins 2019; Kassi 2019).
- Clarification in Introduction and Methods: The Introduction now contains a short, qualitative overview of typical endocrine irAE frequencies to orient the reader. The Methods section explicitly notes that heterogeneity in screening protocols and event definitions precluded a formal pooled prevalence meta-analysis for individual endocrinopathies.
These modifications directly address the reviewer’s concern, align the text with data quality constraints, and enhance scientific rigor while preserving the integrity of the prognostic analyses (OS and PFS).